# Supercritical CO$_2$-assisted rapid synthesis of covalent organic framework-based electrocatalyst for efficient two-electron oxygen reduction reaction

Junqi Song[1,4], Zhiqiang Zhang[2,4], Weiping Li[3,4], Chunli Liu[3], Guodong Feng [3], Yaqiong Su [3], Kai Xi[3] ✉, Hong Yi [2] ✉, Changhai Yi[1] & Lan Peng[1] ✉

Covalent organic frameworks (COFs) hold significant promise as electrocatalysts, but their synthesis is typically constrained by prolonged reaction times (>72 h), high temperatures (>120 °C), and the use of organic solvents. Conventional methods also involve multiple freeze-pump-thaw cycles, complicating scalability. Herein, we report a supercritical carbon dioxide (Sc-CO$_2$)-assisted strategy for the rapid synthesis of COFs, enabling their direct in-situ growth on carbon substrates. This supercritical-solvothermal approach yields COF@CNT composites that exhibit effective electrocatalytic performance towards the two-electron oxygen reduction reaction (2e$^-$ ORR). The resulting catalysts achieve a H$_2$O$_2$ production rate of 94 mol g$_{cat}^{-1}$ h$^{-1}$ and a Faradaic efficiency exceeding 95% at 800 mA cm$^{-2}$. By reducing the consumption of organic solvents, shortening reaction durations, and circumventing high temperatures, this method provides a scalable and efficient route for COF synthesis. Overall, the Sc-CO$_2$ strategy provides a promising platform for the rapid development of COF-based electrocatalysts, combining enhanced efficiency, scalability, and environmental compatibility.

Hydrogen peroxide (H$_2$O$_2$) is a crucial chemical widely used across industrial processes[1–3], including disinfection, bleaching, and environmental remediation. Traditionally, H$_2$O$_2$ is produced via the energy-intensive anthraquinone method[4,5], which involves complex separation steps and significant environmental impact. Recently, the two-electron oxygen reduction reaction (2e$^-$ ORR) has emerged as a more sustainable approach for on-site H$_2$O$_2$ generation[6,7]. However, existing electrocatalysts, including noble metals[8,9], transition metals[10,11], and carbon materials[2,12], each face inherent limitations[13]. Noble metal

catalysts exhibit high activity but suffer from scarcity and prohibitive costs. Transition metal catalysts, while they are more abundant, tend to favour four-electron pathway, yielding water instead of H$_2$O$_2$, and thus compromising product selectivity. Metal-free carbon catalysts offer their high surface areas and tunable properties but typically lack well-defined active sites, limiting their catalytic precision and efficiency.

Covalent organic frameworks (COFs) have emerged as a potential solution to these challenges. Composed of light elements, COFs offer

[1]Technology Institute, National Engineering Laboratory for Advanced Yarn and Fabric Formation and Clean Production, Wuhan Textile University, Wuhan, Hubei, PR China. [2]The Institute for Advanced Studies (IAS), Wuhan University, Wuhan, Hubei, PR China. [3]School of Chemistry, Engineering Research Center of Energy Storage Materials and Devices, Ministry of Education, National Innovation Platform (Center) for Industry-Education Integration of Energy Storage Technology, State Key Laboratory of Electrical Insulation and Power Equipment, Engineering Research Center of Energy Storage Material and Chemistry, Xi'an Jiaotong University, Xi'an, Shaanxi, PR China. [4]These authors contributed equally: Junqi Song, Zhiqiang Zhang, Weiping Li. ✉e-mail: kx210.cam@xjtu.edu.cn; hong.yi@whu.edu.cn; lpeng@wtu.edu.cn

high surface areas and long-range order[14,15], enabling the modular design of active sites[16,17]. However, their poor intrinsic electrical conductivity[18,19] has hindered their electrocatalytic performance. To address this, strategies such as heteroatom doping[20,21], post-synthetic functionalization[22], linkage modification[23,24], and metallation[25] have been employed[26]. For instance, incorporating transition metals into a porphyrin-based COF has been shown to improve ORR activity, with cobalt coordination enhancing performance compared to the metal-free framework[27]. Similarly, multi-step modifications, such as converting reversible imine linkages into stable benzoquinoline units and demethylating methoxy groups to expose phenolic hydroxyls, have been used to increase active site density[28]. Despite these advances, metal-based systems remain complex and have yet to deliver the ideal selectivity for $H_2O_2$ production.

In recent years, the integration of triazine motifs into COF backbones has garnered significant attention. The stability and conjugation of triazine units enhance electron transfer[29,30], as demonstrated by a fully conjugated three-dimensional (3D) COF that achieves over 80% $H_2O_2$ selectivity in alkaline media[31]. Replacing phenyl groups with triazine units has also been shown to lower the ORR overpotential[32]. While theoretical studies suggest that extended conjugation promotes localized electron migration and strengthens interactions between catalytic sites and intermediates[33], the exact mechanism by which conjugation improves $2e^-$ ORR remains unclear. Moreover, COF-based catalysts still struggle to meet practical performance standards, especially under high current densities.

To overcome the electrical conductivity limitations of COFs, composite strategies that combine COFs with conductive materials have been explored[34,35]. For example, vertically anchoring porphyrin-based COF nanosheets onto multiwalled carbon nanotubes (CNT) has significantly enhanced catalytic activity in electrochemical reactions[36]. However, exposed conductive substrates can interfere with $H_2O_2$ selectivity during ORR, highlighting the need for precise control over composite design. Unlike traditional solvothermal methods relying on organic solvents, the supercritical $CO_2$ solvothermal approach replaces organic media with supercritical fluids, boosting mass transfer rates while offering environmental friendliness[37,38]. This method is compatible with industrial supercritical technologies, holding potential for scale-up synthesis[39,40]. Notably, the current laboratory scale has not yet achieved $CO_2$ recycling, whereas specialized large-scale recycling equipment has been developed in industries such as power generation and chemical processing[41,42].

In this work, we introduce a supercritical $CO_2$-assisted approach for the rapid synthesis of metal-free COF electrocatalysts, facilitating efficient $H_2O_2$ production via the $2e^-$ ORR (Fig. 1). By harnessing the unique gas-like diffusivity and liquid-like solvation properties of supercritical $CO_2$ (Sc-$CO_2$), we achieve polymerization within minutes, significantly reducing the reaction times required by conventional solvothermal methods. The resulting alkyne-functionalized COF incorporates both triazine and $sp$-hybridized alkyne units, which optimize the binding energies of oxygen intermediates and enhance $H_2O_2$ selectivity. Furthermore, the in-situ growth of COFs on CNT improves both electrical conductivity and mass transport. This approach achieves an $H_2O_2$ production rate of 94 mol $g_{cat}^{-1}$ $h^{-1}$ and a Faradaic efficiency exceeding 95% at a current density of 800 mA cm$^{-2}$. Our strategy provides valuable insights into the rational design of high-performance $2e^-$ ORR systems and highlights the transformative potential of Sc-$CO_2$ technology in advancing electrocatalytic materials for sustainable chemical synthesis.

## Results

### Minutes-level COF-based catalyst synthesis and characterization via supercritical solvothermal strategy

We present a supercritical solvothermal method for the rapid synthesis of COFs within minutes (Fig. 2a and Supplementary Fig. 1). In this approach, $CO_2$ is introduced into the reactor under precisely controlled temperature and pressure, reaching supercritical conditions that enable fine-tuned process control[43,44]. Unlike conventional organic solvents, supercritical fluids offer superior permeability and diffusivity, accelerating monomer polymerization and facilitating reversible bond formation[45]. This significantly reduces reaction times from days to minutes compared to traditional solvothermal methods[43,46].

As a proof of concept, three triazine-based COFs−COF$_{TSA}$, COF$_{TAZ}$, and COF$_{Ph}$−were synthesized via supercritical solvothermal method (SC-COFs) in 1 h and organic solvothermal method (OS-COFs) in 3 days, respectively (Fig. 2e). Fourier-transform infrared (FTIR) spectra confirmed imine bond formation, with a distinct C=N stretching vibration at 1624 cm$^{-1}$, while the absence of -NH$_2$ and -C=O signals verified complete condensation (Supplementary Fig. 6). SC-COFs and OS-COFs have similar morphology (Supplementary Fig. 7).

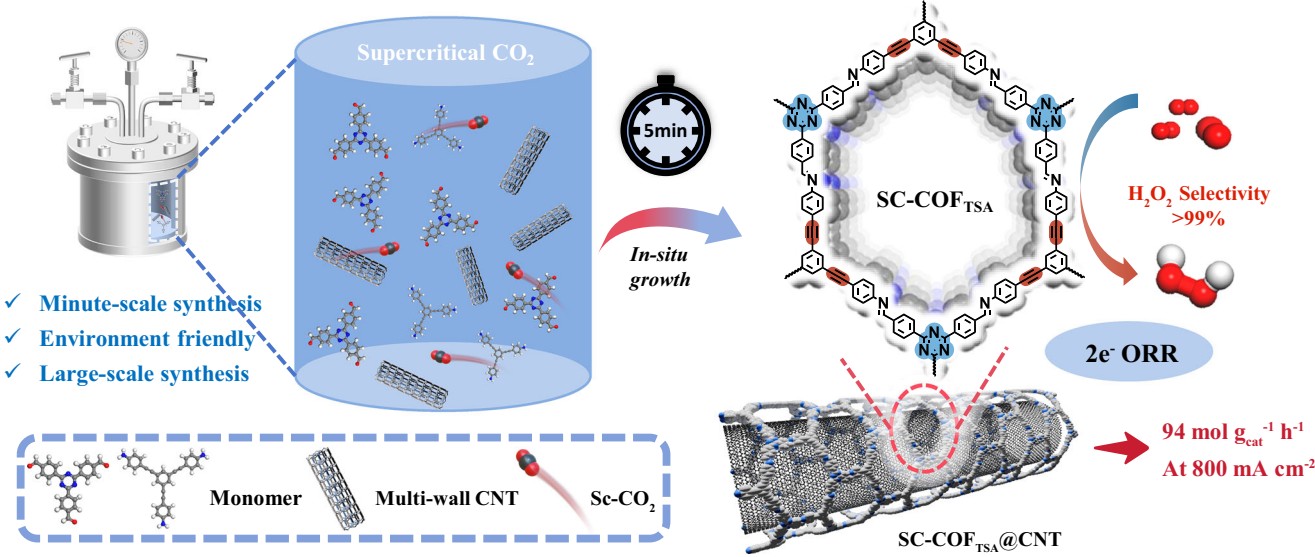

**Fig. 1 | Schematic illustration of the supercritical $CO_2$-assisted solvothermal synthesis of ORR electrocatalysts.** A conceptual depiction of the rapid and environmentally benign synthesis process, where supercritical $CO_2$ facilitates in-situ polymerization and direct growth of covalent organic frameworks on carbon nanotube substrates to yield high-performance catalysts for the two-electron oxygen reduction reaction.

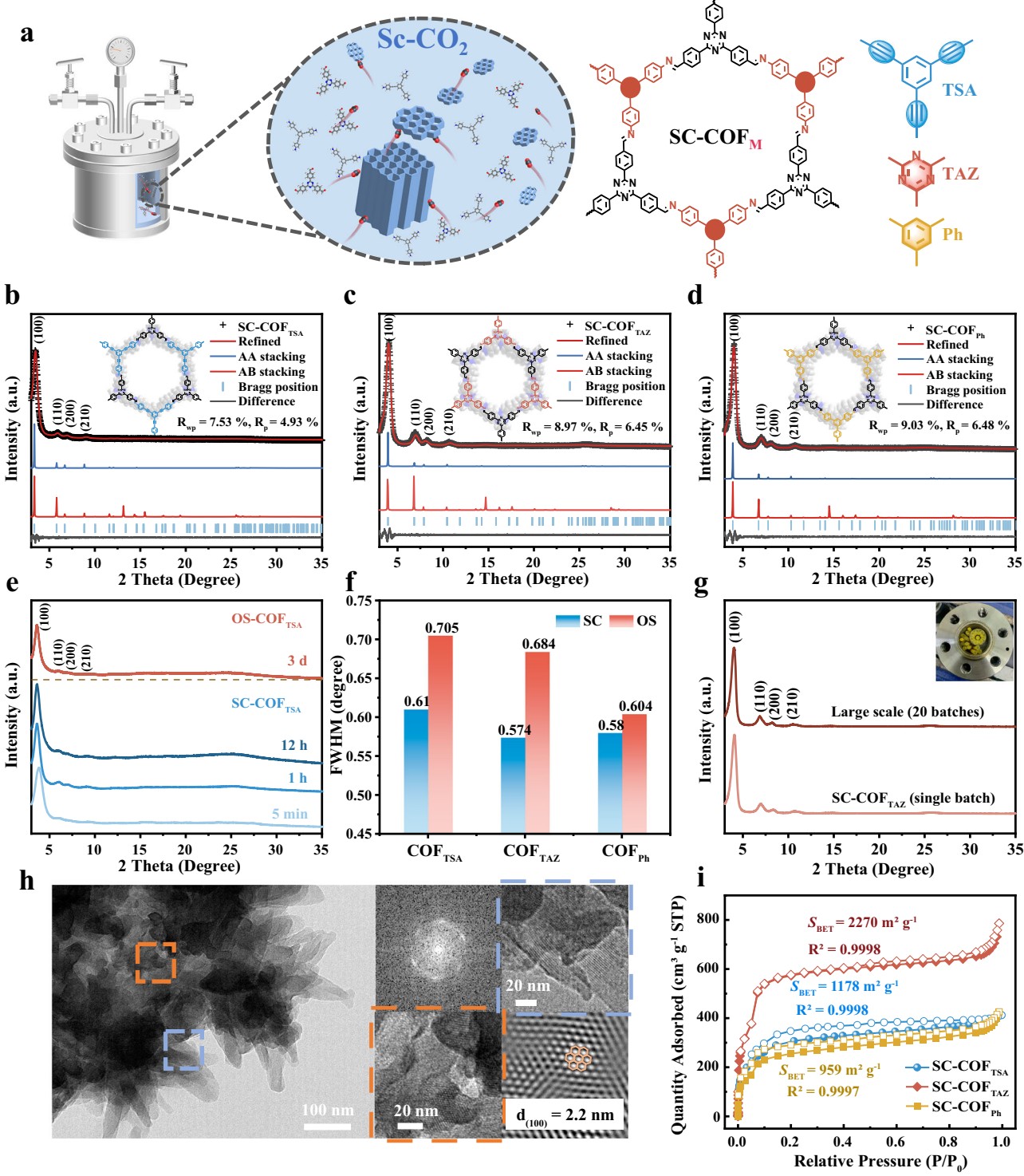

**Fig. 2 | Structure and characterization of SC-COFs. a** Schematic illustration of SC-COF$_M$ synthesis in Sc-CO$_2$. Powder X-ray diffraction (PXRD) pattern of **b** SC-COF$_{TSA}$, **c** SC-COF$_{TAZ}$ and **d** SC-COF$_{Ph}$. **e** PXRD pattern of COF$_{TSA}$ synthesized within minutes via the Sc-CO$_2$ solvothermal method. **f** Full width at half maximum (FWHM) values of (100) crystal plane peak for the as-synthesized COFs. **g** PXRD pattern of large-scale-synthesized SC-COF$_{TAZ}$ prepared over 20 batches (inset: corresponding digital photograph). **h** High-resolution transmission electron microscope (HR-TEM) image of SC-COF$_{TAZ}$. **i** Nitrogen adsorption-desorption isotherms of SC-COF.

The crystal structures of triazine-based COFs are resolved by using powder X-ray diffraction (PXRD) measurements, Pawley refinements and the structural simulations are performed in Materials Studio. In the PXRD pattern, the peak signals at 3.43°, 3.98°, and 4.01° are assigned to the (100) facets for COF$_{TSA}$, COF$_{TAZ}$, and COF$_{Ph}$, respectively (Fig. 2b–d). All three COFs synthesized by supercritical solvothermal method and organic solvothermal method show high crystallinity (Supplementary Fig. 5). Structural model optimization and Pawley refinement confirmed an AA stacking arrangement (Supplementary Figs. 2–4 and Supplementary Table 1). Notably, PXRD patterns of COFs synthesized via the supercritical approach in 1 h exhibited narrower full-width at half-maximum (FWHM) values for characteristic

diffraction peaks compared to those of materials obtained after three days under conventional conditions (Fig. 2f). $N_2$ adsorption-desorption measurements reveal Brunauer-Emmett-Teller (BET) surface area of SC-COFs is higher than that of OS-COFs (Supplementary Fig. 8 and Supplementary Table 4), indicating higher crystallinity and more time-saving.

Confirmed by FTIR analyses (Supplementary Fig. 11), COF@CNT composites also can be successfully synthesized by supercritical solvothermal method, and PXRD patterns of COF@CNT composites also show AA stacking motif (Supplementary Fig. 10), indicating that CNT incorporation does not alter the interlayer stacking of COFs. COF@CNT composites synthesized by supercritical solvothermal method show more uniform morphology compared with that synthesized by organic solvothermal method (Supplementary Fig. 12).

We further shorten the synthesis time of SC-COFs and SC-COF@CNT composites from 1 h to 5 min. Yield rate of SC-COFs and SC-COF@CNT composites synthesized in 1 h is higher (Supplementary Table 3). Scanning electron microscopy (SEM) image of SC-COF@CNT composites synthesized in 1 h shows thicker COF layer than that of SC-COF@CNT composites synthesized in 5 min (Supplementary Fig. 13). PXRD patterns confirmed the successful synthesis in 5 min (Supplementary Figs. 9 and 15). $N_2$ adsorption-desorption measurements reveal SC-COFs and SC-COF@CNT composites synthesized in 5 min have comparable BET surface area with that synthesized in 1 h (Supplementary Figs. 14 and 16 and Supplementary Table 2). Leveraging advantages of Sc-CO$_2$, we developed an efficient COF-based catalyst synthesis strategy that achieves high crystallinity within just 5 min (Fig. 1).

Control experiment has been conducted to prove the supercritical fluid influence for the rapid synthesis of COF-based catalyst. With monomers and CNT were reacted in *n*-Butanol with 12 M acetic acid at 80–90 °C for 1 h without supercritical $CO_2$, XRD patterns (Supplementary Fig. 17) show significantly reduced crystallinity, which clearly demonstrates that supercritical $CO_2$ is essential for achieving rapid formation of crystalline COF-based catalysts. Scalable synthesis advantages of the supercritical solvothermal method have also been verified over the traditional solvothermal method. Firstly, we attempted to increase the reaction amount in a single reactor. As shown in Supplementary Fig. 18, when the batches were gradually increased in a single traditional solvothermal reaction vessel, the crystallinity of the products gradually decreased, indicating that the traditional solvothermal synthesis method has strict requirements for reaction ratios. In contrast, when we increased the reaction batches in a single vessel using the supercritical solvothermal method, the products still maintained good crystallinity. Secondly, as depicted in Supplementary Fig. 19, up to 20 batches could be added to this vessel. After a 5-min reaction, both SC-COF and SC-COF@CNT exhibited good crystallinity (Fig. 2g and Supplementary Fig. 20), with single-batch yields reaching approximately 300 mg and 450 mg, respectively. This demonstrates the minute-scale synthesis advantage of the supercritical solvothermal method in large-scale production. Supercritical $CO_2$ technology has been extensively applied in industries such as power generation and chemical processing[41,42,47]. Therefore, this scalable synthesis approach holds great promise for further promoting industrial production. Beyond significantly reducing synthesis time, the Sc-CO$_2$ approach also enhanced surface area, scalability, and material quality, yielding large-scale batches with up to 88% efficiency—surpassing conventional solvothermal methods.

High-resolution transmission electron microscopy (HR-TEM) exhibits that SC-COFs display high crystallinity (Fig. 2h and Supplementary Figs. 21 and 22), which matched well with the AA stacking mode of three COFs. Solid-state $^{13}C$ NMR spectra further corroborated the formation of imine linkages, displaying characteristic peaks near 157 ppm (Supplementary Figs. 23–25). X-ray photoelectron spectroscopy (XPS) tests are further carried out to determine the elemental composition of SC-COFs (Supplementary Fig. 26); no signals of metals can be detected, which proves metal-free features of SC-COFs, which can also be proven by energy-dispersive X-ray spectroscopy (EDS) mapping (Supplementary Fig. 27). $N_2$ adsorption-desorption isotherms at 77 K revealed pronounced microporosity, with SC-COF$_{TAZ}$ exhibiting the highest BET surface area of 2270 m$^2$ g$^{-1}$ (Fig. 2i). BETSI model can also be used to fit and analysis BET surface area of three COFs (Supplementary Figs. 28–33). Quenched solid density functional theory (QSDFT)[48,49] analysis showed well-defined pore sizes centered around 2.52 nm for SC-COF$_{TSA}$ and 2.27 nm for SC-COF$_{TAZ}$ and SC-COF$_{Ph}$ (Supplementary Fig. 14d). Thermogravimetric analysis (TGA) analysis revealed that the mass loss of the materials at 500 °C was below 10% (Supplementary Fig. 34), while the characteristic COF peaks remained consistent after immersion in diverse solvents for one week, demonstrating the thermal and chemical stability of these SC-COFs (Supplementary Fig. 35). FTIR spectra of SC-COF$_{TSA}$ after immersion in 1.0 M KOH for 12 h verified the stability of imine bonds. The characteristic peak of the imine bond at 1624 cm$^{-1}$ remains distinct in SC-COF$_{TSA}$ post-treatment (Supplementary Fig. 75). These results highlight the large surface areas, uniform pore networks, structural tunability, and stability of SC-COFs, reinforcing their potential as robust platforms for electrocatalysis and other applications that demand abundant, accessible active sites.

## Electrocatalytic activity of triazine-based COFs for 2e$^-$ ORR

The electrocatalytic performance of triazine-based COFs toward the ORR was evaluated using a four-electrode rotating ring-disk electrode (RRDE) setup in O$_2$-saturated 0.1 M KOH at 1600 rpm. The collection efficiency of the platinum ring (Pt-ring) was determined via [Fe(CN)$_6$]$^{4-}$/[Fe(CN)$_6$]$^{3-}$ redox measurements (Supplementary Fig. 36). The cyclic voltammetry curve of SC-COF$_{TSA}$ shows more obvious peak current in O$_2$ atmosphere than in Ar atmosphere (Supplementary Fig. 37), indicating strong ORR catalytic activity. Electrocatalytic experiments comparing SC-COF$_{TSA}$ synthesized at 5 min versus 1 h showed that the 1-h product exhibited superior performance stability (Supplementary Fig. 38). To standardize sample preparation and ensure comparable properties across experiments, we prepared all COF-based catalysts with 1-h reaction time. Linear sweep voltammetry (LSV) results (Fig. 3a, b) showed high H$_2$O$_2$ selectivity for all three COFs, with SC-COF$_{TSA}$ achieving the highest selectivity of 99.3%, surpassing SC-COF$_{TAZ}$ (96.8%) and SC-COF$_{Ph}$ (96.3%). These findings highlight the critical role of triazine units in promoting the two-electron (2e$^-$) ORR pathway[50].

Among the three COFs, SC-COF$_{TSA}$ exhibited the lowest Tafel slope (Fig. 3c), suggesting enhanced ORR kinetics. When the LSV curves were normalized by BET surface area (Fig. 3d), SC-COF$_{TSA}$ displayed the highest onset potential, further confirming its superior catalytic performance. At 0 V, the turnover frequency (TOF) values of SC-COF$_{TSA}$ and SC-COF$_{TAZ}$ exhibit close values, which are higher than that of SC-COF$_{Ph}$ (Supplementary Fig. 39). This enhancement is likely attributed to the electron-donating alkyne moieties, which optimize active-site interactions with oxygen intermediates and facilitate charge transport[2,51–53]. Chronoamperometry measurements (Fig. 3e) demonstrated that SC-COF$_{TSA}$ maintained over 95% H$_2$O$_2$ selectivity after 5 h of continuous operation, with minimal loss in disk current density, indicating reliable long-term stability.

## Structural insights and electrocatalytic mechanism of triazine-based COFs in the ORR

The structure-property relationship between the molecular configuration of COFs and their electrocatalytic performance in ORR was elucidated with density-functional theory (DFT) calculations and in-situ attenuated total reflection flourier transform infrared spectroscopy (ATR-FTIR) (Fig. 4, Supplementary Fig. 82, and Supplementary data 1–3). The electrostatic potential (ESP) maps (Supplementary

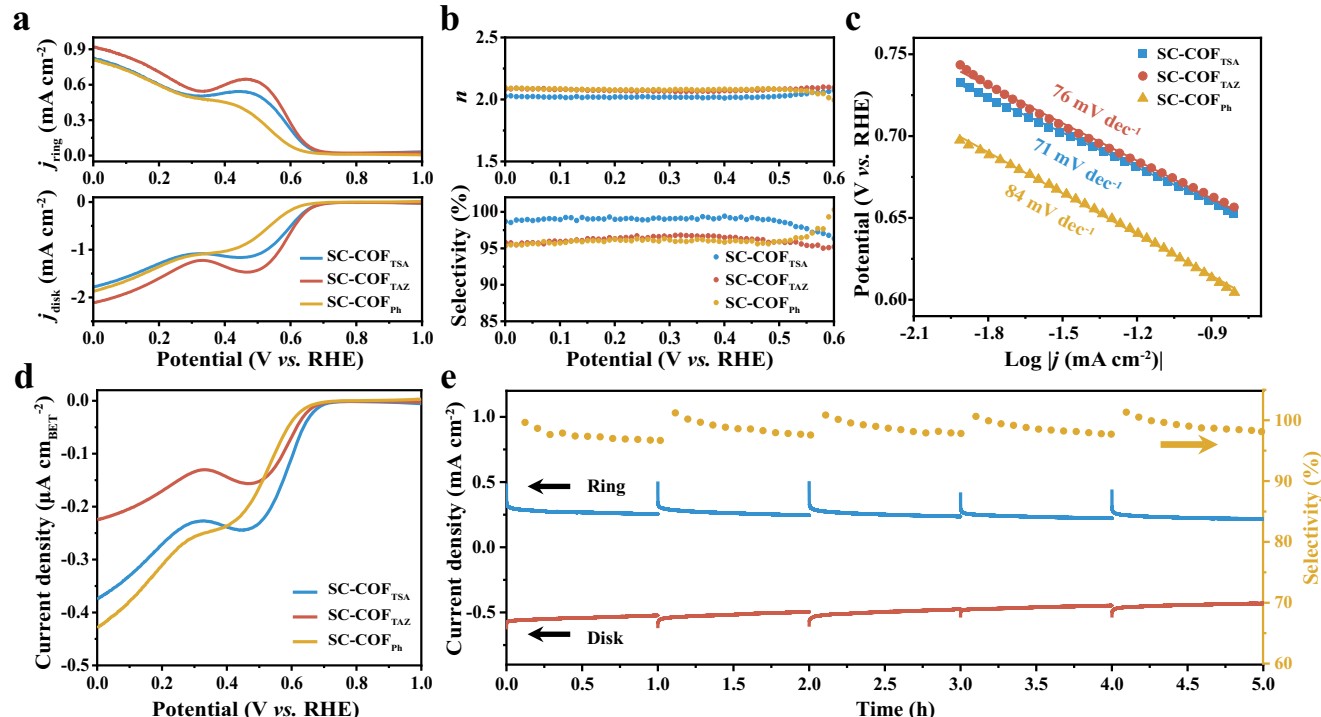

**Fig. 3 | Electrocatalytic performance in $O_2$-saturated 0.1 M KOH electrolyte.**
**a** Linear sweep voltammetry (LSV) curve showing activity trends. **b** $H_2O_2$ selectivity and electron transfer number ($n$). **c** Tafel slopes highlighting kinetic characteristics. **d** LSV curves normalized by BET surface area for intrinsic activity comparison.
**e** Stability measurements at -0.56 V with periodic platinum ring reductions to eliminate $PtO_x$ accumulation.

Fig. 40) of the monomers show higher electron distribution densities on alkyne carbons and triazine nitrogens. Therefore, as revealed by the ESP maps of the COF rings (Fig. 4a), compared to phenyl groups, conjugated alkynyl groups exhibit significantly enhanced electron density, forming an electron-rich environment that favors the dissociation of *OOH intermediates (Supplementary Figs. 46–48). Furthermore, the ESP map of $COF_{TAZ}$ further indicates that the electron-withdrawing nature of the triazine groups reduces the electron cloud density around the triazine ring, leading to a significant lowering of the highest occupied molecular orbital (HOMO) energy level while the lowest unoccupied molecular orbital (LUMO) remains relatively unchanged. This results in $COF_{TAZ}$ having the largest band gap (Fig. 4b and Supplementary Fig. 42). As for $COF_{Ph}$, the phenyl units are electroneutral compared to triazine units, show distinct positive-negative charge separation. Band gap calculations based on DFT were performed for three different covalent organic frameworks (COFs) (Supplementary Fig. 45), among which $COF_{TSA}$ possesses the narrowest band gap, enabling higher electron transport efficiency. This band gap trend was further validated by cyclic voltammetry (CV) (Supplementary Fig. 43) and UV-Visible diffuse reflectance spectroscopy (Supplementary Fig. 44 and Supplementary Table 5), showing a consistent order ($COF_{TSA}$ < $COF_{Ph}$ < $COF_{TAZ}$). This highlights that the electronic structure of COFs can be fine-tuned by modifying structural units. The wettability of COFs also matters to the catalytic performance. The contact angle of $COF_{TSA}$ is the largest (Supplementary Fig. 41), indicating that $COF_{TSA}$ has a higher gas mass transfer efficiency at the catalytic interface and is conducive to enhancing the catalytic stability of the electrode.

Figure 4c and Supplementary Fig. 49 illustrate the full reaction pathway of the ORR. In the 2e⁻ ORR process, $O_2$ is first adsorbed onto the active sites, undergoes electron reduction to form the *OOH intermediate, subsequently captures additional electrons and combines with protons to generate HOOH, and finally desorbs to produce $H_2O_2$. The optimal catalytic sites for $COF_{TSA}$ (site 2), $COF_{TAZ}$ (site 1), and

$COF_{Ph}$ (site 3) were identified by calculating the adsorption energies of different COF sites for *OOH (Supplementary Figs. 46–48). Supplementary Fig. 50 shows that the binding abilities of the three COFs to $O_2$ are comparable. Based on the free energy diagram of the 2e⁻ pathway at an operating potential of 0.7 V, the rate-determining step was determined to be the formation of *OOH, with the overpotentials following the trend: $COF_{TSA}$ (0.37 eV) < $COF_{TAZ}$ (0.65 eV) < $COF_{Ph}$ (1.04 eV) (Fig. 4d). Similarly, the overpotentials for the 4e⁻ ORR at 1.23 V were calculated, and the lower overpotentials indicated higher activity for the 2e⁻ reaction (Supplementary Figs. 51 and 52).

The volcano plot (Fig. 4e), where the Gibbs free energy of *OOH adsorption was selected as the descriptor[52,53]. The selectivity for $H_2O_2$ versus $H_2O$ is determined by the tendency for O–O bond cleavage; the left side of the volcano plot corresponds to stronger *OOH binding affinity, which favors the dissociation of the O–O bond adsorbed on the catalytic sites, thereby dominating the 4e⁻ pathway selectivity. All three COFs designed in this study lie on the right side of the volcano plot, exhibiting weaker *OOH binding affinity and thus higher $H_2O_2$ selectivity. Furthermore, as moving rightward from the volcano apex, the catalytic activity decreases due to the rate limitation of $O_2$-to-*OOH conversion, with $COF_{TSA}$ being the closest to the apex. Collectively, these results confirm that $COF_{TSA}$ exhibits the optimal $H_2O_2$ selectivity and catalytic activity.

These speculations are substantiated through in-situ infrared spectroscopy, with Fig. 4f depicting the spectral changes at different voltages. As the applied potential decreased from 1.0 V to 0.05 V, distinct peaks emerged at 1079, 1245, and 1650 cm⁻¹, corresponding to adsorbed *$O_2^-$, *OOH, and *$H_2O$ species, respectively. The presence of *$O_2^-$ and *OOH at 0.80 V suggests an initial $O_2$ chemisorption step followed by protonation to *OOH, while the *$H_2O$ signal appeared at 0.775 V and intensified at lower potentials, indicating that adsorbed water facilitates proton transfer in *OOH, thereby enhancing ORR kinetics. Additionally, a weak peak at 1393 cm⁻¹ associated with *HOOH confirms the rapid release of $H_2O_2$, aligning with the experimentally

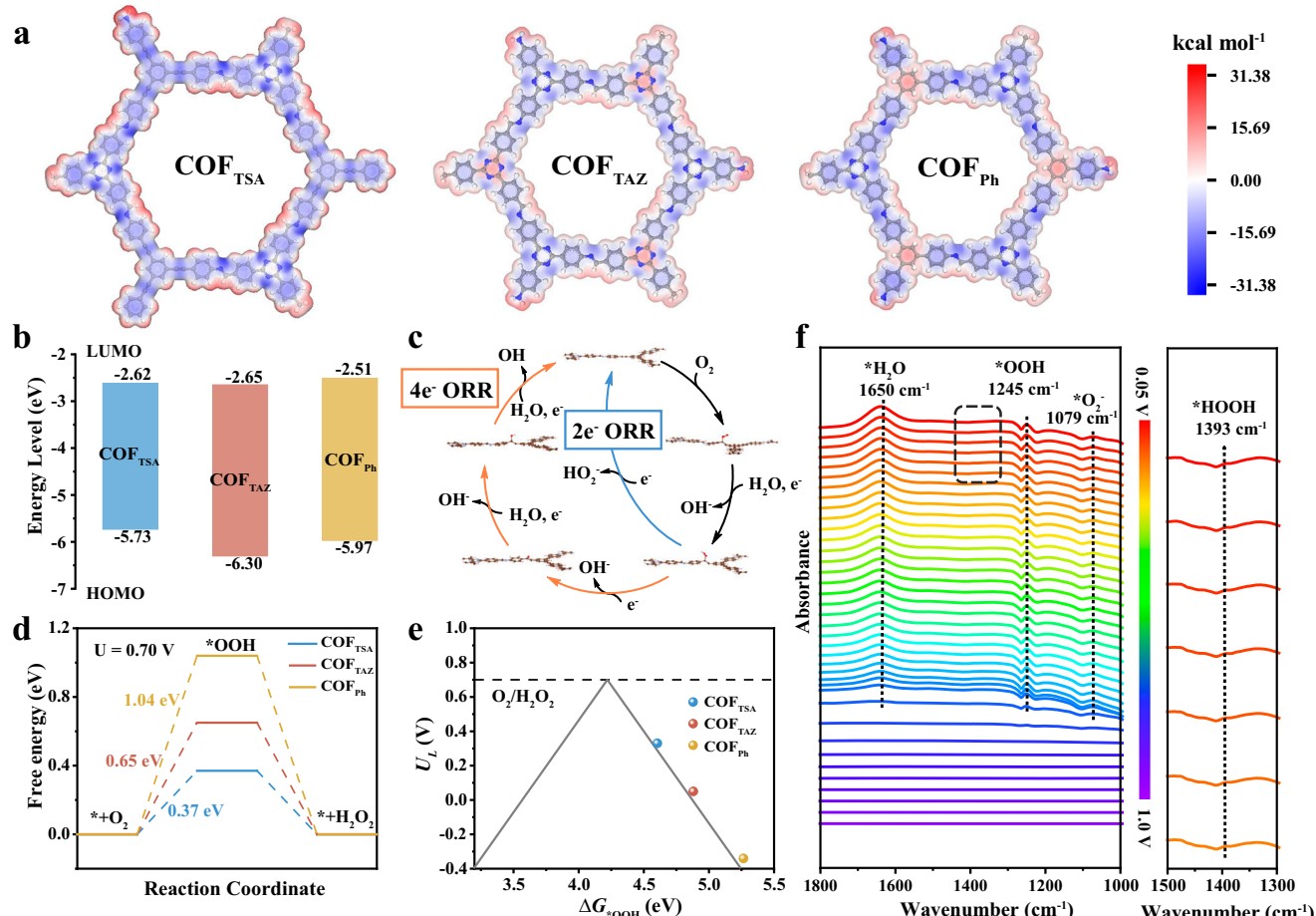

**Fig. 4 | Catalytic mechanism and structural insights for SC-COF_TSA in the ORR.**
**a** Electrostatic potential (ESP) maps of the COF_TSA, COF_TAZ and COF_Ph. **b** LUMO-HOMO of COF_TSA, COF_TAZ and COF_Ph. **c** Reaction mechanism of 2e⁻ ORR and 4e⁻ ORR. **d** Free energy diagrams for the 2e⁻ ORR pathway. **e** Calculated activity-volcano curve for 2e⁻ ORR. **f** In-situ ATR-FTIR spectra of SC-COF_TSA under different potentials.

observed high $H_2O_2$ selectivity. Overall, these results demonstrate that the incorporation of alkyne units enhances both catalytic activity and selectivity, providing molecular-level insights into the ORR mechanism of triazine-based COFs.

## SC-COF@CNT synthesis and electrocatalytic performance for 2e⁻ ORR

Electrocatalysts with hierarchical pore structures can effectively balance mesoporous and microporous domains[54,55], improving electrolyte infiltration and mass transport. However, pure COFs often exhibit predominantly microporous networks and low electrical conductivity[35], limiting their applicability in electrocatalysis. To overcome these limitations, we synthesized SC-COF@CNT composites via the in-situ integration of carbon nanotubes (CNT) with COFs under Sc-$CO_2$ conditions (Fig. 5a and Supplementary Figs. 59 and 70). Compared with the particle structure formed by nanosheet assembly in SC-COF_TSA, the nanofiber structure of SC-COF_TSA@CNT-30 is more conducive to catalytic site exposure and mass transport. (Fig. 5b, c). A similar structural transformation was observed in SC-COF_TAZ@CNT-30 and SC-COF_Ph@CNT-30, demonstrating the versatility of this in-situ composite strategy (Supplementary Figs. 54 and 55). Transmission electron microscopy (TEM) further confirmed that CNT are wrapped with ~20 nm-thick COF layers (Fig. 5c and Supplementary Fig. 56), promoting efficient charge transfer (Fig. 5f). Further verified by XPS and EDS mapping, COF@CNT composites exhibit no metallic signals (Supplementary Figs. 57 and 58). Additionally, the N 1 s spectra display two main peaks at 398.2 eV (C–N in triazine rings) and 401.5 eV (C=N in

imine linkages) (Supplementary Figs. 26c and 57c), demonstrating that CNT introduction does not change the bonding information of COF.

Moreover, the COF layer thickness could be finely tuned by adjusting the CNT-to-COF ratio (Supplementary Figs. 53, 60), enabling optimization of SC-COF_TSA@CNT for the ORR. Notably, increasing CNT content optimizes the onset potential of ORR while maintaining high $H_2O_2$ selectivity (Supplementary Fig. 61). However, excessive CNT loading (50 wt%) reduced selectivity, with 30 wt% identified as the optimal composition. Electrochemical tests revealed that bare CNT exhibited poor $H_2O_2$ selectivity (<75% at 0 V), whereas SC-COF_TSA@CNT-30 maintained over 96% selectivity across a wide potential range (Supplementary Fig. 62). Although its $H_2O_2$ selectivity was approximately 1.3% lower than SC-COF_TSA, the composite displayed significantly higher onset potential, thus accelerating the 2e⁻ ORR kinetics (Fig. 5d and Supplementary Fig. 63). LSV curves in 0.1 M $H_2O_2$ Ar-saturated solution (Fig. 5e) revealed minimal $H_2O_2$ decomposition currents for both SC-COF_TSA@CNT-30 and SC-COF_TSA, underscoring the catalysts' promise for $H_2O_2$ electrosynthesis with minimal side reactions.

To investigate the role of CNT, BET surface area analyses showed comparable surface areas and pore distributions for SC-COFs and SC-COFs@CNT (Supplementary Figs. 14 and 16). SC-COFs@CNT exhibited a lower charge transfer resistance ($R_{ct}$) (Fig. 5f), which further decreased with increasing CNT content (Supplementary Fig. 64). Double-layer capacitance ($C_{dl}$) measurements (Fig. 5g and Supplementary Figs. 65 and 66) revealed a larger electrochemical surface area for SC-COF_TSA@CNT than for

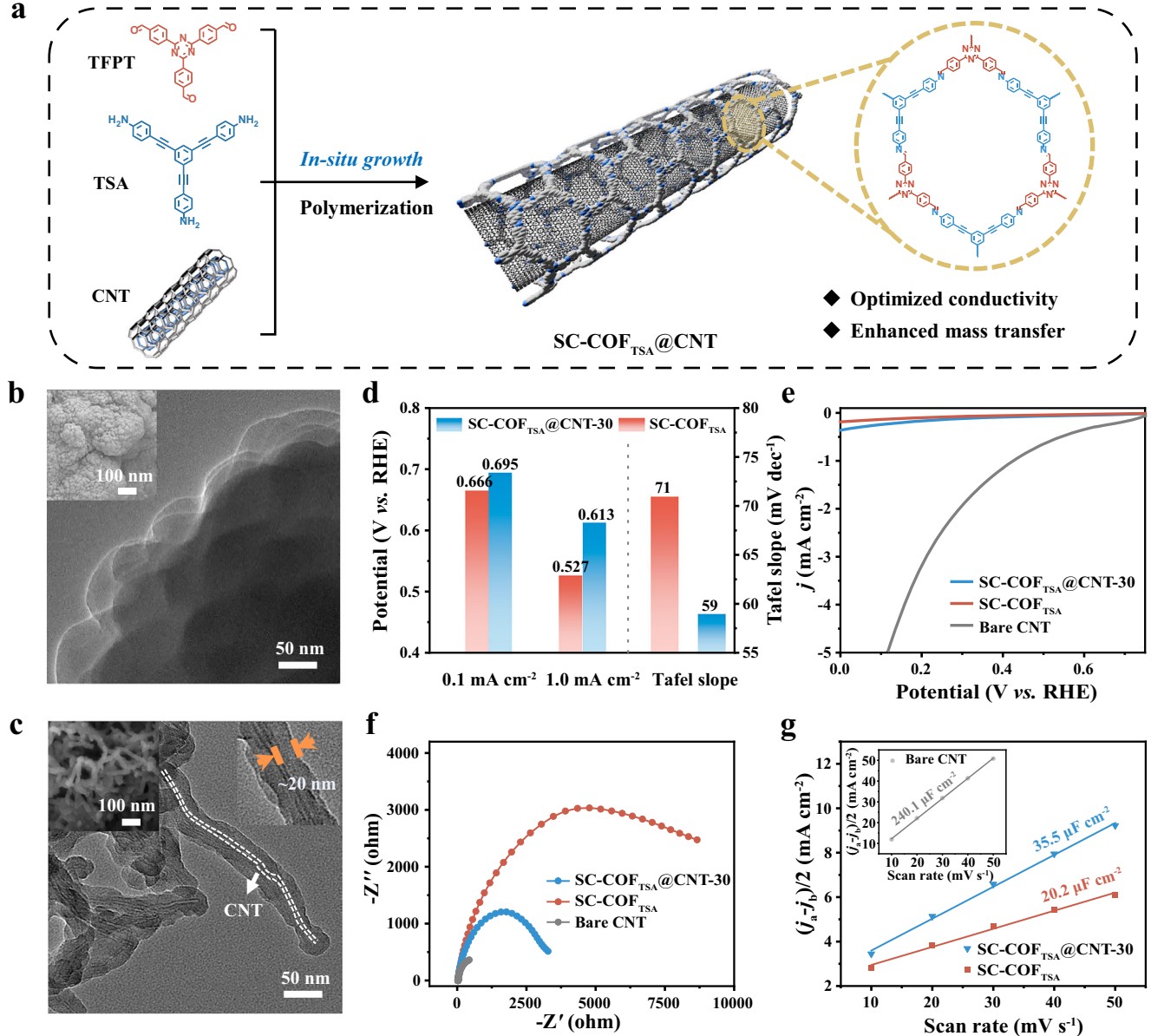

**Fig. 5 | Structural and electrochemical characterization of SC-COF@CNT materials. a** Schematic diagram of in-situ COF growth on CNT. **b** TEM image of SC-COF$_{TSA}$ (the SEM image was in the top left corner). **c** TEM image of SC-COF$_{TSA}$@CNT-30 (the SEM image was in the top left corner and the TEM for the thickness of COF was in the top right corner). **d** Comparison of onset potential and Tafel slope. **e** LSV curves in Ar-saturated 0.1 M KOH containing 0.1 M H$_2$O$_2$. **f** Electrochemical impedance spectroscopy (EIS). **g** Fitted double-layer capacitance (C$_{dl}$).

SC-COF$_{TSA}$, suggesting an increase in exposed active sites due to CNT incorporation. Similar enhancements in electrochemical properties were observed for SC-COF$_{TAZ}$@CNT-30 and SC-COF$_{Ph}$@CNT-30 (Supplementary Figs. 67 and 68), reinforcing the effectiveness of this in-situ composite strategy. Among the three SC-COFs@CNT, SC-COF$_{TSA}$@CNT exhibited the highest ORR activity (Supplementary Fig. 69), highlighting the beneficial role of the alkyne functional group in improving catalytic performance. Compared with the reported catalysts, SC-COF$_{TSA}$ demonstrates selectivity that is comparable to that of certain metal-containing catalysts (Supplementary Fig. 78).

### Evaluation of SC-COF$_{TSA}$@CNT in a flow electrolyzer
To investigate the practical viability of SC-COF$_{TSA}$@CNT-30, we evaluated its ORR performance and stability under high current densities using a flow electrolyzer (Fig. 6a and Supplementary Fig. 80). The catalyst achieved a current density of 400 mA cm$^{-2}$ at an applied potential of 0.34 V (Fig. 6b) while maintaining a Faradaic efficiency (FE) above 95% for H$_2$O$_2$ production across a broad range of current densities (50–800 mA cm$^{-2}$) (Fig. 6d). At 800 mA cm$^{-2}$, SC-COF$_{TSA}$@CNT-30 sustained a high H$_2$O$_2$ production rate of 94 mol g$_{cat}^{-1}$ h$^{-1}$, with only a minor performance decline attributed to electrolyte leakage (Fig. 6c). In contrast, the FE values of SC-COF$_{TAZ}$@CNT-30 and SC-COF$_{Ph}$@CNT-30 exhibited a decreasing trend with the increase in current density (Supplementary Figs. 72 and 73). At 800 mA cm$^{-2}$, their H$_2$O$_2$ production rates were 88.2 mol g$_{cat}^{-1}$ h$^{-1}$ and 76 mol g$_{cat}^{-1}$ h$^{-1}$, respectively, which were lower than that of SC-COF$_{TSA}$@CNT-30 (Supplementary Fig. 74). These results are consistent with the high selectivity observed in RRDE tests, confirming robust 2e$^-$ ORR activity.

Long-term electrolysis at a constant current of 200 mA cm$^{-2}$ for 8 h resulted in the production of a 0.94 wt% H$_2$O$_2$ solution, with negligible fluctuations in operating potential or FE (Fig. 6e). The

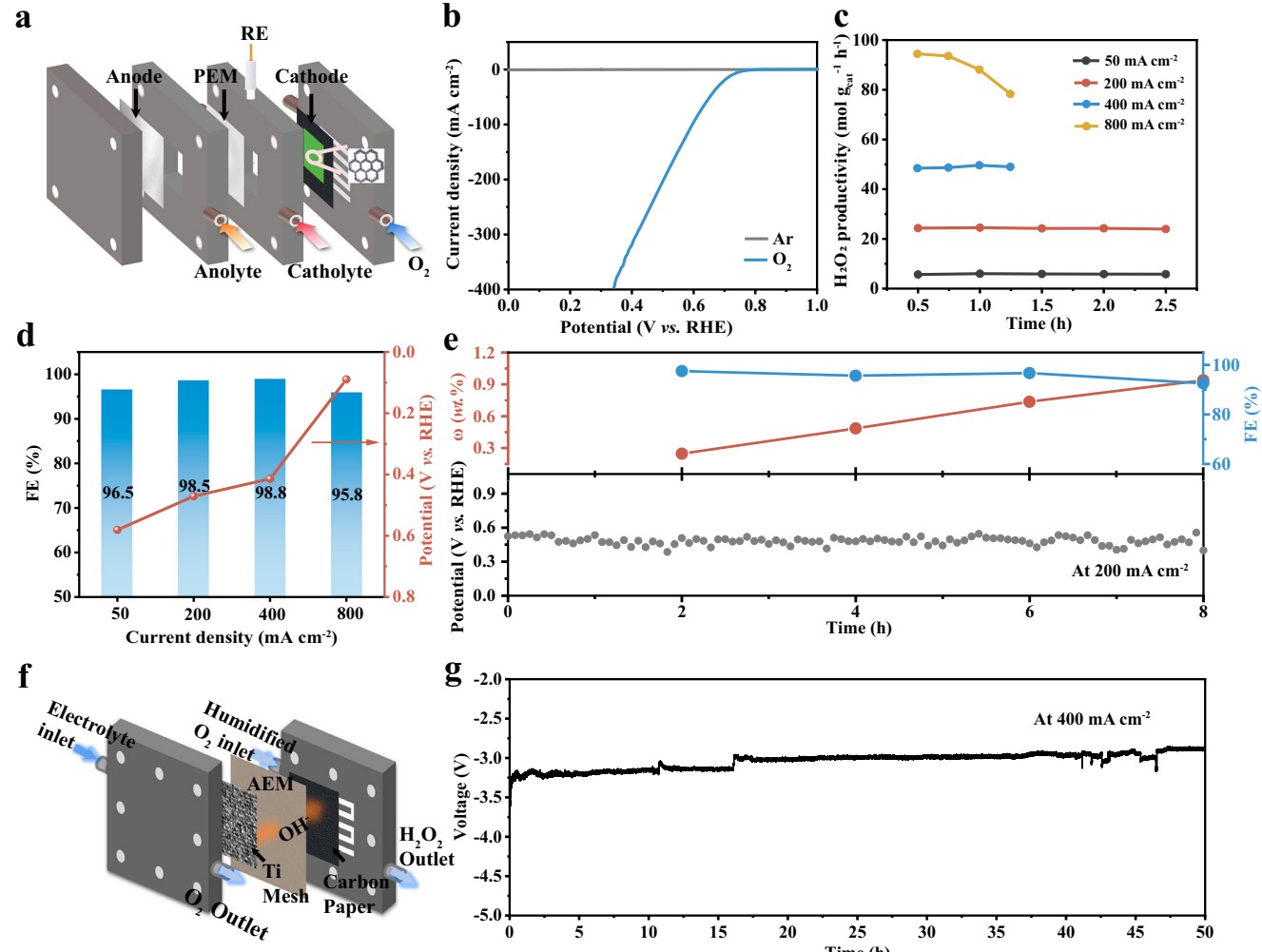

**Fig. 6 | Performance evaluation of SC-COF$_{TSA}$@CNT in flow cells and MEAs.**
**a** Schematic illustration of the gas-diffusion flow cell structure. **b** LSV curves of SC-COF$_{TSA}$@CNT-30 in the flow cell. **c** H$_2$O$_2$ production rates at different current densities. **d** Faradaic efficiency (FE) for H$_2$O$_2$ and corresponding potential across different current densities. **e** Long-term chronopotentiometry (CP) test of the catalyst in the flow cell (Without $i$R). **f** Schematic illustration of the membrane electrode assembly (MEA) structure. **g** Long-term CP test of the catalyst in the MEA configuration (Without $i$R).

catalysts maintain good crystallinity and imine bond stability even after being immersed in a 0.1 M KOH solution for 12 h. Compared with reported electrocatalysts for H$_2$O$_2$ synthesis, SC-COF$_{TSA}$ exhibit H$_2$O$_2$ selectivity comparable to those outstanding materials (under the condition of 0.1 M KOH, 0.2 V vs. RHE), such as COF-CN (95%)[56], O-HATP-HKH-CON (98%)[57], and ZnCo-ZIF-C3 (~100%)[58]. After loading CNT, the TOFs of the three composites are all updated. This phenomenon indicates that the loading of CNTs can effectively optimize the catalytic activity of the COF-based composites. Notably, SC-COF$_{TSA}$@CNT-30 combines high selectivity with superior FE and TOF (Supplementary Figs. 71 and 78 and Supplementary Table 7). A comparative analysis of H$_2$O$_2$ yield (Supplementary Fig. 79) reveals a monotonic increase with current density, with SC-COF$_{TSA}$@CNT-30 delivering a maximum yield of 94 mol g$_{cat}$$^{-1}$ h$^{-1}$ at 800 mA cm$^{-2}$, and its performance in terms of production rate in the low current density range is also observable.

To further assess its commercial potential, SC-COF$_{TSA}$@CNT-30 was incorporated into a membrane electrode assembly (MEA) (Fig. 6f, and Supplementary Fig. 81) and operated continuously for over 50 h at 400 mA cm$^{-2}$ without noticeable degradation (Fig. 6g). The catalyst consistently exhibited robust electrochemical stability under demanding current densities, demonstrating its suitability for long-

term operation. In addition, the structural and electrochemical performance changes of the catalyst before and after the chronopotentiometry (CP) test were further investigated. SEM (Supplementary Fig. 76) analysis demonstrated no appreciable morphological modifications. PXRD analysis (Supplementary Fig. 77a) revealed that the diffraction peak corresponding to the (100) crystal plane remained unchanged after the reaction. FTIR testing (Supplementary Fig. 77b) showed that the characteristic imine bond (C=N) peak at 1624 cm$^{-1}$ remained distinct, indicating no alteration in the dynamic covalent bonds of the COF framework. XPS analysis (Supplementary Fig. 57) confirmed no shift in the imine nitrogen signal at 401.5 eV in the N 1 s spectrum, further supporting that the electronic environment of the C=N bonds remained unaltered. Additionally, LSV curves (Supplementary Fig. 77c) before and after the reaction exhibited strong overlap, demonstrating that the catalytic active sites remained intact without deactivation during prolonged operation. These results establish SC-COF$_{TSA}$@CNT as a promising platform for on-site chemical production, water treatment, and other industrial applications requiring efficient H$_2$O$_2$ generation. Combining high selectivity, durability, and scalability, SC-COF$_{TSA}$@CNT holds significant potential for both research and commercial applications in advanced electrosynthesis.

## Discussion

We introduced a rapid supercritical $CO_2$-assisted solvothermal strategy for synthesizing triazine-based, metal-free COFs, significantly reducing synthesis time compared to conventional methods. The incorporation of alkyne and triazine units optimizes the electronic structure and generates electron-rich active sites, with SC-COF$_{TSA}$ exhibiting enhanced oxygen adsorption, and promoting both $H_2O_2$ selectivity and production rate. In-situ integration of COFs with CNT improves electrical conductivity and mass transport, resulting in increased onset potential while maintaining a Faradaic efficiency exceeding 95% even at high current densities. Electrochemical measurements combined with computational studies reveal that alkyne functionalization facilitates *OOH binding and desorption, favoring the 2e⁻ ORR pathway while effectively suppressing the competing 4e⁻ pathway. Furthermore, the Sc-$CO_2$ approach enables precise control over COF crystallinity and CNT integration, producing catalysts with durability and high performance under practical conditions. Collectively, this work establishes Sc-$CO_2$ processing as a powerful and scalable route for the rapid fabrication of COF and COF@CNT composites, paving the way for scalable and efficient electrocatalytic $H_2O_2$ production.

## Methods

### Protocol of supercritical solvothermal method. (SC-COF)

The supercritical reactor setup comprises a 50 mL stainless-steel pressure vessel, temperature-controlled heating jacket, flow control valves, and real-time monitoring system for pressure and temperature. The operational procedure involves: loading the reaction mixture into the reactor chamber, transferring it to the stainless-steel vessel, pressurizing high-purity $CO_2$ to the target pressure (8 MPa), maintaining isothermal conditions for 5–60 min, and recovering products via controlled decompression.

### Protocol of organic solvothermal synthesis. (OS-COF)

For organic solvothermal polymerization, the protocol involves dissolving monomers in a co-solvent system, introducing a catalytic acid, degassing via freeze-pump-thaw cycles, and performing isothermal heating under vacuum for 3 days.

### Scalable synthesis of COFs based on different solvothermal methods

**Organic solvothermal method.** For a typical single-batch synthesis of OS-COF, the reaction mixture comprises TFPT (9.8 mg, 0.025 mmol) and TSA (10.6 mg, 0.025 mmol) as monomers, 0.5 mL of $n$-butanol and 0.5 mL of $o$-dichlorobenzene as the organic solvent system and 0.1 mL of 6 M acetic acid aqueous solution as the catalyst. Components for different batch quantities are added to a 10 mL Schlenk tube, followed by a 3-day reaction. After polymerization, the products are collected, washed, and characterized by PXRD.

**Supercritical solvothermal method.** For a typical single-batch SC-COF synthesis, the reaction mixture includes TFPT (9.8 mg, 0.025 mmol) and TSA (10.6 mg, 0.025 mmol) as monomers [or TFPT (9.8 mg, 0.025 mmol), TAPT (8.85 mg, 0.025 mmol) as monomers with CNT (5.6 mg) as the substrate], 0.5 mL of $n$-butanol (or 0.2 mL for SC-COF$_{TAZ}$) as the co-solvent, and 0.1 mL of 12 M acetic acid aqueous solution as the catalyst. Reaction vessels (prepared according to the batch quantity) are loaded with the components, placed in a 50 mL stainless-steel autoclave, and reacted for 5 min. Following pressure release, products are collected, washed, and characterized by PXRD.

### Electrochemical characterization of ORR

The ORR activity and $H_2O_2$ selectivity of the catalyst were evaluated using linear sweep voltammetry (LSV) with the rotating ring-disk electrode (RRDE) technique in $O_2$- and Ar-saturated 0.1 M KOH solutions. Electrochemical impedance spectroscopy (EIS), double-layer capacitance ($C_{dl}$), and stability were also measured. High current density performance, including Faradaic efficiency (FE) for $H_2O_2$ and catalytic stability, was assessed using a gas-diffusion flow cell in 1.0 M KOH solution. Long-term stability at high current densities was further tested using a membrane electrode assembly (MEA) setup.

### Electrosynthesis of $H_2O_2$ in flow cell (FL)

The $H_2O_2$ electrosynthesis performance of COF@CNT at high current density was tested using a flow cell. The volume of the cell is 250 mL. Carbon paper (CP, SGL, 28BC) was used as the gas diffusion electrode, and a working electrode was prepared by spraying catalyst ink onto CP (1 cm²) with a catalyst loading of 0.15 mg cm⁻², which was determined by weighing the mass of the electrode before and after spraying. A platinum sheet and Ag/AgCl (saturated KCl, 0.1989 V) were used as counter and reference electrodes, respectively. N117 was utilized as the proton exchange membrane, with a dimension of 2 cm × 2 cm and a thickness of 183 μm. The membrane was pretreated through the following steps: It was treated in 5 wt% $H_2O_2$ solution at 80 °C for 1 h, followed by soaking in deionized water for 30 min. Subsequently, it was treated in 5 wt% dilute sulfuric acid ($H_2SO_4$) solution at 80 °C for 1 h and soaked in deionized water again for 30 min.

1.0 M KOH solution was used as the electrolyte and flowed in the cathode and anode chambers at a flow rate of 10 mL min⁻¹, respectively. Pure $O_2$ was continuously passed into the gas chamber of the cathode at a flow rate of 20 sccm. Quantification of the generated $H_2O_2$ using $Ce(SO_4)_2$:

$$H_2O_2 + 2Ce^{4+} \rightarrow 2Ce^{3+} + 2H^+ + O_2 \tag{1}$$

$$c(H_2O_2) = \frac{c(Ce^{4+}) \times V(Ce^{4+})}{2 \times V(H_2O_2)} \tag{2}$$

Before starting the titration, the electrolyte was first acidified (pH: 1 - 2) using 0.05 M $H_2SO_4$ solution and 1,10-phenanthroline-ferrous complex ion (10 μL, 1 mM) was added as an indicator, after which the titration was carried out using a $Ce(SO_4)_2$ standard solution [$c(Ce^{4+})$: 10 mM] and recorded the volume consumed [$V(Ce^{4+})$].

### Electrosynthesis of $H_2O_2$ in membrane electrode assembly (MEA)

The electrocatalytic stability of the catalysts at higher current densities was tested using MEA. The working electrode and counter electrode were prepared using the same method as the working electrode of the flow cell, the working electrode was a CP loaded with COF@CNT material with a catalyst loading of 0.15 mg cm⁻² and the counter electrode was a titanium felt loaded with $IrO_2$ with a catalyst loading of 0.15 mg cm⁻², both with a working area of 1 cm². FAA-3−50 (Fumasep, FuMA Tech) was used as the anion-exchange membrane, with a dimension of 2 cm × 2 cm and a thickness of 50 μm. It was pretreated by soaking in a 0.1 M KOH solution for 12 h. The anode side was circulated with a 1.0 M KOH solution at a flow rate of 10 mL min⁻¹, while the cathode side was supplied only with humidified $O_2$ gas at a flow rate of 20 sccm.

### Computational methods

During the transition state calculation, Gauss View 6.0 was used for molecular modeling. For structural optimization, Becke's three-parameter exchange functional[59] combined with Lee-Yang-Parr correlation functional[60] (B3LYP) was used to calculate the minimum geometry of the large state, DFT-D3[61] was used to consider the van der Waals (vdW) interactions, and 6−311 G (d, p) basis set was used.

Density functional theory (DFT) based calculations were performed by using the Vienna Ab initio Simulation Package (VASP)[62], which enabled us to achieve the relaxed geometries and total energies.

The projector augmented wave (PAW)[63] method was adopted to describe the nuclei-electron interactions. The Perdew-Burke-Ernzerhof (PBE)[64] functional within the generalized gradient approximation (GGA)[65] was employed to calculate the exchange-correlation energy. The kinetic cutoff energy was set to 420 eV. The convergence criterion for the electronic self-consistent field (SCF) loop was set to $1 \times 10^{-5}$ eV/atom. The atomic structures were optimized until the residual forces below 0.04 eV/Å.

## Data availability

All relevant data that support the findings of this study are presented in the manuscript and supplementary information file. The experimental datasets associated with this work are available at https://doi.org/10.6084/m9.figshare.29985097. Source data are provided with this paper.

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

## Acknowledgements

We are grateful for the financial support provided by Hubei Technological Innovation Program Funding (2025BAB025), the National Natural Science Foundation of China (Grant Nos. 22031008, 22278329 and 92472124), the Guangdong Basic and Applied Basic Research Foundation (No. 2023A1515012260) and the Wuhan Textile University Startup Fund. We thank the Analytical and Testing Center of Wuhan Textile University for providing PXRD and FTIR test. We gratefully acknowledge the National Innovation Platform (Center) for Industry–Education Integration of Energy Storage Technology and the Instrumental Analysis Center of Xi'an Jiaotong University for their support. Supercomputing facilities were provided by Hefei Advanced Computing Center and Computing Center in Xi'an.

## Author contributions

L.P. supervised the project and provided guidance on the project. J.S. and Z.Z. performed the experiments. W.L., C.L., G.F. and K.X. conducted the DFT calculations. J.S., Z.Z., H.Y., K.X., C.Y. and L.P. wrote and revised the paper. Y.S. provided computational software. All authors contributed to the analysis. J.S., Z.Z. and W.L. contributed equally to this work.

## Competing interests

The authors declare no competing interests.
