## [Transparent Peer Review file · Nature Communications]

Supercritical CO₂-Assisted Rapid Synthesis of Covalent Organic Framework-Based Electrocatalyst for Efficient Two-Electron Oxygen Reduction Reaction

Corresponding Author: Professor Hong Yi

This manuscript has been previously reviewed at another journal that is not operating a transparent peer review scheme. The manuscript was considered suitable for publication without further review at Nature Communications.

Version 0:

Reviewer comments:

Reviewer #1

(Remarks to the Author)

This manuscript presents a supercritical CO₂-assisted solvothermal method for the rapid and environmentally benign synthesis of covalent organic frameworks (COFs) for application in two-electron oxygen reduction reaction (2e⁻ ORR) to produce hydrogen peroxide (H₂O₂). The approach enables gram-scale synthesis of highly crystalline COFs within minutes, bypassing traditional energy- and time-intensive solvothermal routes. The resulting SC-COFTSA and its composite SC-COFTSA@CNT-30 demonstrate superior activity and Faradaic efficiency (>95%) for H₂O₂ electrosynthesis, especially under high current densities in both flow cells and MEAs.

The work is experimentally thorough and addresses critical COF synthesis scalability and electrocatalyst design bottlenecks. The combination of structural characterization, in situ spectroscopy, and DFT analysis strengthens the mechanistic rationale. The manuscript can be considered for publication in Nature Communications after addressing the major concerns listed below.

1. While the authors emphasize a synthetic breakthrough with their supercritical CO₂-assisted approach, certain practical aspects require further clarification. Specifically, The manuscript claims that the Sc-CO₂ method operates under milder and more scalable conditions than conventional solvothermal synthesis. However, both approaches utilize similar conditions, namely, 8 MPa pressure, 80 °C temperature, n-butanol as a solvent, and 12 M acetic acid as a catalyst. Thus, the practical advantage of the Sc-CO₂ system in terms of temperature and pressure is not fully evident.

2. The schematic in the main manuscript indicates that COF formation occurs within 5 minutes. However, the experimental section in the Supporting Information describes a reaction duration of 1 hour, followed by Soxhlet extraction in THF at 90 °C for 6 hours and vacuum drying at 80 °C for 12 hours, procedures that are also required for conventionally synthesized COFs. This raises questions regarding the actual time savings and workflow simplification attributed to the Sc-CO₂ approach. Given these points, the authors are requested to clarify:

- There is a discrepancy between the stated 5-minute synthesis time and the 1-hour reaction period in the ESI. Whether the extensive post-synthetic purification steps are essential in both methods and how this affects the claimed “rapid synthesis” advantage.
- The specific benefits of the Sc-CO₂ methodology in terms of scalability, safety, and cost relative to conventional solvothermal methods. Addressing these discrepancies would greatly strengthen the credibility of the claimed advantages and help readers assess the real-world applicability of the proposed method.

3. Is CO₂ recycled in the system? A brief discussion on the industrial feasibility or techno-economic outlook would strengthen the manuscript.

4. It is strongly recommended that the authors perform a control experiment in which the monomers are mixed in n-butanol with 12 M acetic acid aqueous solution as the catalyst, followed by 5 minutes of sonication and heating at 80–90 °C for 1 hour without the use of supercritical CO₂. This would clarify whether COF formation can occur under these relatively simple solvothermal conditions. If COF formation is possible under such conditions (i.e., without freeze-pump-thaw cycles or supercritical CO₂), it could represent a less complex and more easily scalable synthesis protocol. Demonstrating that COFs do not form under these conditions would, conversely, provide stronger justification for the necessity of the Sc-CO₂-assisted approach. Either outcome would significantly enhance the scientific rigor and practical relevance of the study.

5. It is important to clarify whether the authors observe any significant differences in crystallinity or porosity between COFs synthesized via Sc-CO₂ vs conventional solvothermal methods beyond time savings.
6. Are there notable differences in COF properties (e.g., stacking, defect density) when grown in the presence of CNTs? Detailed characterization discussion should be included in the manuscript.
7. To convincingly demonstrate that the synthesized catalysts, particularly the MWCNT composite materials, are truly metal-free, it is recommended to include X-ray photoelectron spectroscopy (XPS) analysis. This would provide direct evidence regarding the absence of residual metal impurities, which is critical when claiming a completely metal-free catalytic system. Such data would significantly strengthen the credibility of the electrocatalytic performance results and the work's overall impact.
8. It is recommended to estimate the electrochemical band gap of the COFs via CV and compare it with the optical band gap from solid-state UV-vis and the DFT-calculated values. This would provide a more comprehensive validation of the electronic structure and support the mechanistic insights.
9. The authors support their claim of enhanced H₂O₂ selectivity via DFT and in situ ATR-FTIR spectroscopy. However, further discussion is warranted on how the OOH binding energy (0.56 eV for SC-COFTSA) mechanistically leads to suppressed 4e⁻ ORR. Can the authors provide additional insight into the kinetic barrier for O–O bond cleavage vs desorption in these systems?
10. Although stability over 50 hours at 400 mA cm⁻² in MEA is impressive, more information is needed: How stable is the structural integrity (e.g., PXRD, FTIR, XPS) post-operation? Can the catalyst be regenerated or reused without activity loss?
11. The authors are encouraged to revise the Abstract, Discussion, and Conclusion sections to eliminate redundant phrasing (e.g., repeated use of “efficient H₂O₂ production”) for improved clarity and conciseness. Additionally, please carefully check the manuscript and Supporting Information for typographical and stylistic errors—for example, “Insuit” should be corrected to “In situ” in Figures 1 and 5. Ensure consistent and proper formatting of scientific units throughout the text (e.g., spacing and superscripts).

Reviewer #2

(Remarks to the Author)

In the manuscript "Supercritical CO₂-Assisted Rapid Synthesis of Covalent Organic Frameworks for High-Performance 2e⁻ Electrocatalytic Oxygen" contributed by Peng et al, the authors introduce a novel method for rapidly synthesizing triazine-based, metal-free covalent organic frameworks (COFs) using supercritical carbon dioxide (Sc-CO₂)-assisted solvothermal processing. They demonstrate the excellent electrocatalytic performance of these COFs in the oxygen reduction reaction (ORR). Additionally, the in-situ integration of COFs with carbon nanotubes (CNT) further enhances conductivity and mass transport efficiency, achieving high Faradaic efficiency at high current densities. Despite the improved performance, the overall content of the manuscript suffers from significant omissions and deficiencies that severely impact the completeness and persuasiveness of the research. For example, the research on COF preparation using supercritical carbon dioxide is not a novel methodology, as extensive studies have been conducted on diverse COF synthesis approaches. The article's title emphasizes hydrogen peroxide-based COF preparation, yet the actual investigation centers on the catalytic performance of COFs supported by carbon nanotubes, resulting in a mismatch between the title and the core research content. Furthermore, the three distinct COF monomers selected by the authors lack explicit comparative analysis, which fails to effectively highlight structure-activity relationships. Therefore, the manuscript is not suitable for publication in Nature Communications due to the following issues.

Other related issues have also been raised

Q1. The manuscript describes the synthesis of three triazine-based COF materials (SC-COFTSA, SC-COFTAZ, and SC-COFPh), but the rationale behind the structural design and its impact on catalytic performance remains inadequately addressed. Specifically, the influence of alkyne functionalization and varying aromatic units on the catalytic behavior is not thoroughly discussed. Moreover, the role of SC-COFTAZ as a control group is not clearly articulated, leaving readers uncertain about its purpose in the study.

Q2. The manuscript lacks a detailed analysis of the COF crystal materials, including essential information such as the crystallographic parameters (a, b, c) and the space group. These details are crucial for understanding the structural characteristics and properties of the materials.

Q3. During the refinement of SC-COFTSA in Figure 2f, the diffraction peaks obtained from AA and AB stacking simulations show a certain degree of similarity to the peak shapes from the experimental X-ray diffraction data, which does not conform to the expected peak shapes in the simulation. To ensure the accuracy and reliability of the simulation results, it is recommended that the authors re-run the simulation to guarantee the accuracy and reliability of the data.

Q4. The peak position of C=N mentioned in the article is 1623 cm⁻¹, while the peak position shown in Figure 2d is 1624 cm⁻¹. There is inconsistency between the two, please correct this issue to ensure consistency between the textual description and the data in the figure.

Q5. PXRD indicates that the material possesses crystallinity, necessitating further confirmation through high-resolution transmission electron microscopy (HRTEM) images, which are presently absent from the manuscript.

Q6. The manuscript presents essential characterization data (XRD, FT-IR, SEM) to elucidate the structure and morphology of the materials. However, the analysis lacks depth in terms of surface chemical properties and electronic structure. Incorporating X-ray photoelectron spectroscopy (XPS) or similar techniques would provide critical insights into surface chemistry and electronic properties, enabling a more thorough evaluation of the catalytic performance.

Q7. The manuscript demonstrates the synthesis of three specific COF materials (SC-COFTSA, SC-COFTAZ, and SC-COFPh) using a supercritical CO₂-assisted method. However, the generalizability of this synthesis approach to other COF systems remains unexplored. The lack of validation on a broader range of COF materials raises questions about the universality and applicability of the method.

Q8. Although the article mentions electrochemical measurements, it lacks a detailed description of key electrochemical parameters, which are crucial for in-depth analysis of the experimental results.

Q9. The calculation of the Turnover Frequency (TOF) values is required of three specific COF materials (SC-COFTSA, SC-COFTAZ, and SC-COFPh).

Q10. The experimental data on hydrogen peroxide production rates for the other two materials (SC-COFTSA@CNT-30 and SC-COFPh@CNT-30) are not presented in the article. As a result, it is difficult to discern the performance improvements of these materials.

Q11. The article has pointed out that at a current density of 800 mA cm⁻², the yield reaches 94 mol g cat⁻¹ h⁻¹. From Figure 6c, it can be observed that the yield exhibits a corresponding increasing trend with the elevation of current density. However, a question worth delving into is whether the yield will continue to increase if the current density is further increased. Is it meaningful to continuously elevate the current density for yield testing? What is the optimal current density?

Q12. The paper only calculated the adsorption energy of OOH*, but did not calculate the free energy changes of the entire 2e⁻ ORR reaction pathway. Although OOH* adsorption energy is important, it cannot fully reflect the kinetics and thermodynamic processes of the 2e⁻ ORR reaction. In addition, the active sites for the 2e⁻ ORR reaction were not identified in the paper, and the contributions of different sites to the reaction performance were not calculated. This makes it difficult for readers to understand the source of catalytic performance.

Reviewer #3

(Remarks to the Author)

This work reports the COFs for 2e electrocatalytic oxygen reduction, the revisions are suggested as follows:

(1) Please explain why the C=N bond is marked differently in the infrared images of Figure 2d in the text and Figure S2-4 in SI

(2) The thermogravimetric data of the three COFs and the stability data under different solvent conditions must be provided in order to prove that COF has good thermal and chemical stability.

(3) The infrared images of SC-COFTSA and SC-COFTSA@CNT-30 after immersion in 1.0 M KOH solution for 12 hours should be supplemented to prove the existence of the imine bond.

Reviewer #4

(Remarks to the Author)

In this work the authors report the Sc-CO₂ assisted synthesis of COFs and COF-CNT hybrids. The compounds were employed as ORR electrocatalysts with excellent results. Despite the overall work is interesting and I consider it a cutting-edge research, there is some major concerns I would like to address before publication.

1) For the statement: "Leveraging these properties, we establish a novel COF synthesis strategy that achieves high crystallinity within just five minutes (Figs. 2a, 2c) It should be mentioned that the ScCO₂-assisted synthesis of COF has been already reported before by the corresponding author: <https://doi.org/10.1038/s41467-021-24842-x>. It is true that in their original work they didn't manage to obtain COF/SWCNT hybrid materials but "novel COF synthesis strategy" is still a controversial statement. In addition, the hybridization of COFs with CNTs has been already explored in this field which reduces the novelty of the reported work. (e.g, <https://doi.org/10.1016/j.cej.2022.138062>)

2) The method displays great linker-tolerance as it was demonstrated along the study. This is one of the most strength points of the work because some methods only work for certain systems (e.g. mechanochemistry). To give more value to the versatility of the ScCO₂ synthesis, N₂ sorption isotherms at 77 and the resulting BET area should be contrasted with the solvothermal-synthesis for a fair comparison (at least the reported ones in literature for each system). In addition, COF's community is starting to employ "BETSI" free Python software to analyze the BET surface areas <https://doi.org/10.1002/adma.202201502>. I recommend calculating the BET areas using this software and reporting the adjustments to the BET model for each product (at least the linear regression). Note: ChatGPT is helpful during the installation of the software if the authors have any problems with the process. Literally it can install it by copying and pasting the CMD dialogue.

3) In addition to points (1 and 2): It is true that a crystalline phase can be obtained in 5 minutes as it was demonstrated in Figure S1. However, all the products are reported to be obtained in periods of one hour in the Experimental section. Despite achieving highly crystalline COFs in one hour is still impressive, the statement and Figure 1 are incorrect. To be allowed to ensure the "just five minutes" PXRD vs time, BET vs time, and yield vs time should be plotted for all the products reported.

4) I don't agree with the statement: "Additionally, conventional COF synthesis often involves harsh conditions high temperatures, pressures, and long reaction times in organic solvents complicating the development of green, scalable production methods"

The authors are still employed:

-organic harm solvents (THF for washing the products)

-high temperatures (80°C for ScCO₂ reaction and Soxhlet extraction vs 120°C for solvothermal synthesis is not a big difference)

-complicated set-up (To the best of my knowledge, ScCO₂ set-up is more complicated than conventional solvothermal one)

-and high pressures (8 MPa is relatively high pressure).

Thus, I recommend reformulating or deleting this part.

5) The author claims achieving "gram-scale" synthesis but the synthetic protocols for the scaled-up reactions are not reported. In addition, for the COFs community, achieving the gram-scale synthesis is an important milestone since most of the reactions fail during the scale-up. Thus, the COFs scientific community usually displays photos of vials with the amount of COF per batch. Despite it is still not the best way to demonstrate the "gram-scale synthesis" is the only way to probe it.

Thus, the authors should depict at least two of the gram-scale synthesis (with and without the CNTs) of the COF powders on a scale without vial and provide the corresponding analysis of those batches (PXRD, BET and FTIR) and comparing it with the small synthesis. Otherwise, I would recommend removing the "gram-scale synthesis" from the text.

6) P3 L104. Instead of stating high crystallinity I suggest reporting the FWHM values for the (100) diffraction maxima.

7) For the solid state ^{13}C NMRs: Remaining aldehyde signals around 190 ppm suggest impure phases, not complete polymerizations or non-removed aldehyde linkers (Figure S5, and S7). Also signals around 70 ppm look like impurities (maybe not-removed solvents or C-alpha of amine groups). Could the authors double-check these experiments?

8) Which model of the QSDFT for the pore size distributions calculation was applied? Please include this in the methods section. In addition, the authors should provide the full distributions to demonstrate the absence of mesopores greater than 3 nm (if possible).

9) I have to congratulate the authors for their excellent description of the electrochemical set-up along the supporting information. Thank you very much for explaining accurately the electrochemical experiments.

10) If possible, the authors should calculate the TOF values and compare them with the recent metal-free and COF based compounds.

11) For the mechanistic calculations. Fig 4e depicts the adsorbed O species in the COF models. However, there is no apparent geometry change for COFTAZ and COFPh. Could the authors provide a closer look? In addition, the benzene units of COFPh display no dihedral angles. Considering intermolecular interactions are also important for the electrocatalytic process. Does the authors consider this in their in silico experiments?

12) In Figure 5h, the selectivity of all the COFs summarized in Table S1 should be included for a fair comparison. Despite the selectivity of the catalyst is excellent there are some compounds with really close values. In addition, all the works cited in Table S1 should be mentioned, discussed and cited along the main text to give those works the deserved recognition.

13) Some data from Table S1 is missing. Please revise the cited works and complete the table. In addition, the synthetic protocol for the crystallization of each catalysts is worth mentioning.

14) What is the difference between COF-based materials and COF-based composite materials. Most of the works reported uses Carbon SuperP or related conductive additives so the segregation has no sense. In addition, I've checked the ESI of some of the COF-based materials and for example JUC-660 uses Carbon Black or BUCT-COF-7 uses carbon nanotubes as conductive additives. I'm sorry but the reported classification is not consistent, and it must be reconsidered. I recommend just segregating by metallized catalysts and metal-free catalysts

15) A clear discussion with the other COF-based electrocatalysts mentioned in table s1 regarding activity and selectivity would be beneficial for the article. Right now, the authors claim that their compound outperforms most of the reported catalysts but that statement is too vague. The authors should discuss in which aspects their catalyst could be highlighted and compare it with at least catalysts showing close activity/selectivity.

16) The authors should also report and compare TOF values.

17) Page 8 Line 247 "Post-electrolysis characterization (Figs. S46-48) also confirmed the structural and electrochemical stability of the catalyst." This claim is not supported by Figure S48. The authors are not recording the COF PXRD pattern. Please delete this graph and change the statement.

18) The method for carrying out the Pawley refinements should be mentioned in the ESI.

19) The ScCO₂ reactor should be described in the ESI and if possible a photograph should be included.

Version 1:

Reviewer comments:

Reviewer #1

(Remarks to the Author)

The authors have addressed my comments and therefore I recommend its acceptance in the current form.

Reviewer #2

(Remarks to the Author)

The authors have addressed all the issues, therefore I recommend publishing without change.

Reviewer #3

(Remarks to the Author)

This manuscript presents a supercritical CO₂-assisted solvothermal approach for the rapid and environmentally benign synthesis of covalent organic frameworks (COFs). This approach enables the efficient production of hydrogen peroxide via the two-electron oxygen reduction reaction (2e⁻-ORR). Leveraging the unique gas-like diffusivity and liquid-like solvation properties of supercritical CO₂, polymerisation can be completed in minutes, which is much faster than conventional solvothermal methods. Furthermore, the in situ growth of COFs on carbon nanotubes improves electrical conductivity and mass transfer efficiency.

The revisions are suggested as follows:

(1) For the rapid synthesis of COF, whether the pressure and flow rate of CO₂ have an effect on the crystallinity and synthesis time of COF.

(2) What are the advantages of supercritical CO₂-assisted rapid synthesis compared to other green synthesis methods (e.g. microwave-assisted, ultrasound-assisted, etc.)

Reviewer #4

(Remarks to the Author)

Thank you to the authors for thoroughly addressing the points previously raised. Before the manuscript can be considered for final acceptance, I would like to request a few minor revisions.

1) Original question: The author claims achieving “gram-scale” synthesis but the synthetic protocols ...

Response: Thanks for your comment. We have supplemented large-scale synthesis... ..corresponding to product masses of 294.8 mg and 426.8 mg per batch. PXRD patterns of the large-scale products show identical diffraction peaks....

New comment: I acknowledge the authors for carrying out this experiment. However, their results do not validate the as-called “gram-scale synthesis”. Please, delete this terminology appearing 3 times in the main article: P7, L 187; P5, L 158 and Fig.1 and in the ESI: P19, L 394; P20 L 398.

2) Original question: For the solid state ¹³C NMRs: Remaining aldehyde signals around 190 ppm suggest impure phases, not complete polymerizations or non-removed aldehyde linkers (Figure S5, and S7). Also signals around 70 ppm look like impurities (maybe not-removed solvents or C-alpha of amine groups). Could the authors double-check these experiments?

Response: Thanks for your comment. To address the concerns regarding potential impurities and incomplete polymerization, we have re-conducted SS...

New comment: Despite repeating the experiments, the team presents the NMR data ranging from 200 ppm and 70 ppm. The authors should present the full ss-¹³C-NMR and assign every signals to validate the obtainment of the proposed structures.

3) Original comment: In Figure 5h, the selectivity of all the COFs summarized in Table S1 should be included for a fair comparison. Despite the selectivity of the catalyst is excellent there are some compounds with really close values. In addition, all the works cited in Table S1 should be mentioned, discussed and cited along the main text to give those works the deserved recognition.

Response: Thanks for your comment. To ensure a fair comparative analysis, we have updated Fig. 5h to include the H₂O₂ selectivity data for most COF-based catalysts summarized in Table S6, allowing direct visualization of performance...

New comment: Some of the COF showed on the table and not depicted in figure 5h.

Version 2:

Reviewer comments:

Reviewer #4

(Remarks to the Author)

The authors have addressed the raised points and the article is now ready to publish in the Journal. Thank you very much for all the efforts and congratulations.

made.
